# Hessian-aware Training for Enhancing DNN Resilience to Bitwise Corruptions in Their Parameters

**Tahmid Hasan Pranto**                                    *prantot@oregonstate.edu*
*School of Computer Science*
*Oregon State University*
**Seijoon Kim**                                    *seijoon.kim@samsung.com*
*Samsung Advanced Institute of Technology (SAIT)*
**Lizhong Chen**                                    *lizhong.chen@oregonstate.edu*
*School of Computer Science*
*Oregon State University*
**Sanghyun Hong**                                    *sanghyun.hong@oregonstate.edu*
*School of Computer Science*
*Oregon State University*

**Reviewed on OpenReview:** *https://openreview.net/forum?id=XxlQF4muso*

## Abstract

Deep neural networks are not resilient to parameter corruptions: even a single-bitwise error in their parameters in memory can cause an accuracy drop of over 10%, and in the worst-cases, up to 99%. This susceptibility poses great challenges in deploying models on computing platforms, where adversaries can induce random/targeted bit-flips, e.g., through software-induced fault attacks like Rowhammer. Most prior work addresses this issue with hardware or system-level approaches, such as integrating additional hardware components to verify a model's integrity at inference. However, these methods have not been widely deployed as they require infrastructure or platform-wide modifications.

In this paper, we propose a new approach to addressing this issue: training models to be more resilient to bitwise corruptions to their parameters. Our approach, Hessian-aware training, promotes models to learn *flatter* loss surfaces. We show that existing training methods designed to improve generalization (e.g., through sharpness-aware minimization) do not improve resilience to parameter corruptions. In contrast, models trained with our method demonstrate improved resilience to parameter corruptions, particularly with a 20–50% reduction in the number of bits whose individual flipping leads to a 90–100% accuracy drop. We also characterize the factors that may influence this increased resilience. Moreover, we show the synergy between ours and existing hardware and system-level defenses.

## 1 Introduction

Deep neural networks (DNNs) are *not* resilient to parameter corruptions. Prior work has shown that adversaries, who are capable of causing random/targeted bitwise errors on the memory representation of their parameters, can exploit this property to induce undesirable behaviors. This includes substantial accuracy drop (Hong et al., 2019; Rakin et al., 2019; Yao et al., 2020a), targeted misclassification (Bai et al., 2023; Cai et al., 2021; Rakin et al., 2021a), and backdoor injections (Chen et al., 2021; Cai et al., 2024; Rakin et al., 2020). Recent work have also demonstrated privacy risks, such as model extraction (Rakin et al., 2022).

Most prior work addresses this vulnerability by developing defenses at the hardware- or system-level (Bennett et al., 2021; Rakin et al., 2021b; Li et al., 2021; Di Dio et al., 2023; Zhou et al., 2023a;b; Liu et al., 2023; Wang et al., 2023). While having demonstrated their effectiveness, they are often difficult to implement in practice as they require additional hardware components or system software updates, necessitating infrastructure-wide changes. Moreover, relatively little attention has been given to defenses that make models inherently resilient.

In this work, we study a novel approach that has not been investigated in prior work: enhancing a model's natural resilience to parameter corruptions. By decreasing the number of model parameters whose bitwise corruptions can cause a substantial accuracy drop, or by minimizing the performance degradation resulting from bitwise errors on those parameters, our models remain resilient even in scenarios where hardware- or system-level defenses are not deployable. Moreover, when combined with these existing defenses, the models lower space and computational complexities by reducing the number of parameters to protect.

**Contributions.** We *first* present Hessian-aware training, an adaptation of stochastic Hessian-trace regularization designed to minimize the sharpness of a DNN's loss landscape, thereby making the model less sensitive to parameter variations and more resilient to bitwise errors in its parameters. Prior work on curvature-aware and robustness-oriented methods, such as SAM (Foret et al., 2021), AdaHessian (Yao et al., 2021b), HERO (Yang et al., 2022), and Lipschitz-based approaches (Gouk et al., 2021), leverages loss sharpness, Hessian information, or input-sensitivity constraints to improve generalization, optimization efficiency, or robustness to bounded perturbations. Our empirical analysis shows that curvature-aware methods such as SAM, AdaHessian, and HERO are insufficient to provide robustness against bit-flip corruption. In contrast, Lipschitz-based methods operate in the input space and are not designed to address discrete parameter-level faults induced by bit-flip errors. Our training algorithm addresses this key problem, and we further propose strategies to make the training process computationally tractable when training ImageNet-scale models.

*Second*, we conduct a comprehensive evaluation of our approach across multiple datasets and network architectures, including those commonly used in prior studies. We demonstrate that our training algorithm significantly enhances a model's resilience to bitwise errors to its parameters. Models trained with our approach have 20–25% of fewer parameters where a single bitwise corruption causes a substantial accuracy drop compared to the baseline models. Against multiple targeted bitwise corruptions, our models require $\sim 3\times$ more bit-flips to achieve the same malicious objectives. Moreover, all models trained with our approach achieve accuracy the same as that of the baseline models.

*Third*, we conduct an in-depth analysis of the increased resilience achieved by our approach. Our analysis of visualized loss landscapes across layers shows a great reduction in sharpness, particularly in the layers close to the output. Accordingly, we observe that the number of parameters where a single-bit corruption can cause a significant accuracy drop is mostly reduced in the fully-connected, classification layers. Moreover, the numerical changes in parameter values required to cause the significant accuracy drop is increased.

*Fourth*, we evaluate the compatibility of models trained with our training algorithm with existing defenses. Our results demonstrate great synergy between our approach and prior work's defenses. Fewer parameters are needed to be protected to achieve the same level of resilience observed when the baseline models are used. Moreover, the defenses exhibit reduced runtime and storage overhead with our models.

## 2 Background and Related Work

### 2.1 IEEE-754 32-bit Floating-Point Representation

The IEEE-754 standard defines a 32-bit single-precision floating-point number format, which is widely used to represent DNN parameters in memory. This format employs exponential notation and consists of three components: a sign bit that determines whether the number is positive or negative, the exponent (8 bits) that encodes the scale of the number using a biased representation, and the mantissa (23 bits) that encodes the precision. The significance of these components varies, with the exponent bits–particularly the most significant bits (MSBs)–having a disproportionately large impact on the represented value. For example, flipping the MSB of the exponent can cause drastic changes, such as turning a small value (e.g., 0.002) into an enormously large one (e.g., $6.8 \times 10^{35}$). This characteristic makes protecting MSBs critical for ensuring numerical stability and resilience in DNNs. The majority of bits whose flipping causes a substantial accuracy drop are MSBs. In contrast, flipping bits in the mantissa typically results in minor perturbations with negligible impact on model performance.

## 2.2 Relation to Prior Work

Prior work on curvature-aware and second-order methods, such as SAM (Foret et al., 2021) and AdaHessian (Yao et al., 2021b), primarily aims to improve generalization by leveraging loss sharpness and Hessian information. Contrarily, HERO (Yang et al., 2022) investigates hessian-eigenvalue regularization for improved quantization robustness. Lipschitz-based methods (Gouk et al., 2021) improve robustness by constraining sensitivity to input perturbations. Our analysis in section 3.4 shows input-space or generalization robustness do not directly translate to robustness against bit-flip corruption.

## 2.3 Bit-Flip Attacks on DNNs via Rowhammer

**Rowhammer Vulnerability and Defenses.** Rowhammer is a software-induced hardware fault-injection that exploits the physical structure of DRAM to induce bit-flips in memory (Kim et al., 2014). By repeatedly accessing ("hammering") specific memory rows, an attacker induces electrical disturbances in neighboring rows, causing bit-flips (Di Dio et al., 2023). Rowhammer attacks have evolved from simple single-sided approaches to more sophisticated techniques like double-sided hammering (Gruss et al., 2018), one-location hammering, and remote attacks via GPUs or network interfaces (Konoth et al., 2018), making them a versatile and persistent threat. Existing defenses against Rowhammer primarily rely on hardware or system-level mechanisms, such as proactive memory refreshing, row isolation, or error-correction techniques (Kim et al., 2014; Bennett et al., 2021; Di Dio et al., 2023). These approaches often require modifications to hardware or system software, making them difficult to deploy in practice, particularly in large-scale DNN deployment scenarios.

**Bit-Flip Attacks on DNNs.**

Bit-flip compromises data values in memory, such as DNN parameters, leading to severe dependability issues at runtime. Most bit-flip attacks on DNNs have been demonstrated using Rowhammer, where attackers are even capable of targeting specific bits to flip (Razavi et al., 2016). In DNNs, flipping critical bits, e.g., the MSBs of parameters, can lead to catastrophic changes to their behaviors, such as drastically reducing performance. Prior work has demonstrated that such bit-flip attacks can lead to significant accuracy degradation, targeted misclassification, and backdoor behaviors in DNNs (Rakin et al., 2021a; Yao et al., 2020a; Bai et al., 2023). Existing defenses span hardware-level protections, system-level mechanisms, and algorithmic approaches that aim to reduce model sensitivity (Li et al., 2021; Liu et al., 2023; Wang et al., 2023). However, these defenses often incur significant overhead or provide limited protection against targeted bit-level perturbations. In contrast, our work focuses on enhancing the inherent resilience of DNNs through training, reducing their susceptibility to parameter-level corruptions.

# 3 Our Hessian-aware Training

Now we present our training algorithm to enhance a model's resilience to parameter corruptions. We first study objectives that quantify a model's *sensitivity* to parameter value variations and use them as a loss function to minimize the sensitivity during training. Suppose that a model $f$ uses a loss function $\mathcal{L}$. The rate at which the loss changes in a specific variation direction $v$ within the parameter space can be expressed as the second-order derivative $\partial^2 \mathcal{L} / \partial v^2$. This value encodes how sensitive a model's performance will be when its parameter values are changed along the direction of $v$. Intuitively, a sharp loss landscape has high curvature, where small parameter perturbations can cause large changes in loss, making the model vulnerable to bit-flip errors. In contrast, a smooth landscape has low curvature and is less sensitive to such perturbations. Therefore, minimizing this rate of change across *all* possible directions $v$ makes the model more resilient to parameter corruption.

## 3.1 The Hessian Trace as a Sensitivity Metric

The next question becomes which metrics to use for capturing the sensitivity from the second-order derivatives. A naive approach would compute the magnitude of the second-order derivatives for a sufficiently large number of directions $v$ and average those values. However, even with modern deep-learning frameworks like PyTorch,

which accelerate derivative computations, computing numerous second-order derivatives of the loss at each mini-batch throughout training remains computationally intractable.

Prior work has proposed various approaches to approximating the second-order derivatives (Jiang et al., 2020; Mulayoff & Michaeli, 2020; Li et al., 2018; Keskar et al., 2017; Neyshabur et al., 2017). *Our work utilizes the Hessian trace*, defined as the sum of the eigenvalues of the Hessian matrix, which serves as a tractable proxy for the curvature of the loss landscape. Unlike the full Hessian spectrum, which is costly to compute, there have been efficient methods approximate to compute the trace, such as the Hutchinson's method (Bekas et al., 2007) which requires only a few Hessian–vector products rather than full eigen-decomposition.

The Hessian trace provides a meaningful indicator of model sensitivity to parameter perturbations. A higher trace reflects sharper regions of the loss surface, where small changes in parameters (e.g., bit-flips) are more likely to cause drastic performance degradation. On the other hand, a lower trace corresponds to flatter regions, which have been associated with improved robustness to perturbations and better generalization.

## 3.2 Hessian-aware Training

Our novel approach to minimizing a model's sensitivity to parameter corruption is to explicitly minimize the Hessian trace during training. By encouraging flatter loss surfaces, models trained with our method obtain inherent resilience to parameter corruptions without requiring additional defensive mechanisms, while also complementing and strengthening existing defenses when combined with them.

### 3.2.1 Challenges in Using the Hessian Trace for Resilience

**Naïvely minimizing the Hessian trace does not improve resilience.** Most prior studies have focused on the second-order metrics to characterize the relationship between the *sharpness* of the loss landscape and model *generalization*. However, their relevance to resilience against bitwise parameter corruptions remain unclear. A few closely related works suggest that minimizing the Hessian trace improve resilience to quantization—that induces bounded and relatively *small* perturbations to parameters (Dong et al., 2019; 2020; Yao et al., 2021a; Yang et al., 2022). In contrast, our evaluation of HERO (Yang et al., 2022), the most recent Hessian trace-based training method, shows that such training does not improve resilience to either single-bit or targeted multi-bit corruptions. This indicates that simply minimizing them is insufficient; a careful study is needed to establish how such metrics can be used to strengthen model resilience.

**Minimizing the Hessian trace during training remains computationally expensive.** The Hessian trace makes it feasible to capture second-order characteristics, yet computing it is still costly. At each training iteration, one must choose the number of eigenvalues ($p$), which is equivalent to the number of random probe directions ($v$) in Hutchinson's method (Hutchinson, 1989). In principle, $p$ can be as large as the total number of model parameters (millions or even billions in modern DNNs). Increasing $p$ yields better resilience but incurs exponentially higher computational cost. In designing our algorithm, we empirically choose the value of $p$ that yield the lowest sharpness while keeping the computation manageable.

**Training instability from second-order optimization.** Foret et al. (2021) observed that plugging-in the Hessian trace approximation as an objective into standard numerical optimizers, such as mini-batch stochastic gradient descent (SGD), can introduce instability during training.

### 3.2.2 Minimizing the Hessian Trace in Training

We present our training algorithm, designed to address the three aforementioned challenges. Our approach is to minimize Top-$p$ Hessian trace during training via an additional regularization term, outlined in Algorithm 1. The algorithm extends the popular numerical optimizer, mini-batch SGD. Notably, any gradient-based training methods can be adapted to our Hessian-aware training. The changes we made are highlighted in blue.

---

**Algorithm 1** The Hessian-aware Training

---

**Input:** A model $f$, Training data $D$, Training steps $T$, Learning rate $\eta$, Number of approximation steps $p$, Regularization coefficient $\alpha$

**Output:** A trained model $f_\theta$

1: Initialize $\theta_0$
2: Initialize $\tau$ to 0
3: **for** $t = 1, 2, ..., T$ **do**
4:     Draw a mini-batch $S_t$ from $D$
5:     Compute the loss $\mathcal{L}_{xe}(S_t; f_{\theta_t})$
6:     $Tr_t, \lambda_t \leftarrow 0, \phi$
7:     **for** $i = 1, 2, ..., p$ **do**
8:         Draw a random vector $v_i$
9:         Compute the gradient $g_i$ of the loss $\mathcal{L}_{xe}$
10:        Compute the Hessian matrix $H_i$ along $v_i$
11:        Compute their eigenvalues $\lambda_i$ and trace $Tr_i$
12:        $Tr_t, \lambda_t \leftarrow Tr_t + Tr_i, \lambda_t + \lambda_i$
13:     **end for**
14:     $Tr_t, \lambda_t \leftarrow (1/p)Tr_t, (1/p)\lambda_t$
15:     **if** $\text{Median}(\lambda_t) > \tau$ **then**
16:        $\mathcal{L}_{tot} \leftarrow \mathcal{L}_{xe}(S_t; f_{\theta_t}) + \alpha * Tr_t$
17:     **else**
18:        $\mathcal{L}_{tot} \leftarrow \mathcal{L}_{xe}(S_t; f_{\theta_t})$
19:        $\tau \leftarrow \text{Median}(\lambda_t)$
20:     **end if**
21:     Compute the gradient $g_t$ of $\mathcal{L}_{tot}$
22:     $\theta_{t+1} \leftarrow \theta_t + \eta \cdot g_t$
23: **end for**
24: **return** a trained model $f_\theta$

---

**(line 3–5, 20–21)** We compute the loss $\mathcal{L}$ of a model $f_{\theta_t}$ and update the model parameters $\theta_t$ with its gradient $g_t$. This step is the same as the original mini-batch SGD.

**(line 6–13)** In this step, we compute the Hessian trace and eigenvalues with respect to the model parameters $\theta_t$. Since computing the full Hessian is computationally expensive; we approximate them using a single step of the Hutchinson's method, following the technique used in prior work (Yao et al., 2020b; 2021b).

Suppose the Hessian $H \in \mathbb{R}^{d \times d}$ and random vector $v \in \mathbb{R}^d$ satisfying $\mathbb{E}[vv^T] = \boldsymbol{I}$. $v$ is drawn from Rademacher distribution which ensures half of the discrete probabilities are positive and the other half is negative ($P(v = \pm 1) = 1/2$). $d$ denotes the total number of parameters. In Hutchinson's method, the Hessian trace over a set of random vectors is:

$$Tr(H) = \mathbb{E}[v^T H v] = \frac{1}{p} \sum_{i=1}^{p} v_i^T H v_i$$

where $p$ is the number of random vectors used to approximate. We can obtain $v^T H v$ by computing the gradient of the loss function $\mathcal{L}$ twice as follows:

$$v^T H v = v^T \cdot \frac{\partial}{\partial \theta} \left( \frac{\partial \mathcal{L}}{\partial \theta} \right) \cdot v$$

We follow the prior work (Yao et al., 2020b) to compute set of Top-$p$ eigenvalues $\lambda_p$ as follows:

$$\lambda_p = \frac{v_i^T H v_i}{\|v_i^T\|} \quad \text{for } i = 1, 2, \cdots, p$$

**(line 14–19)** In our experiments, we find that minimizing the Hessian trace computed on all eigenvalues (equal to the total number of parameters) renders the training process computationally intractable as well as

making the optimization process unstable. To address these issues, we first take the $p$-largest eigenvalues to compute the loss. There will be negligible impact since the eigenvalues consist of a few large values (representing the sharpest directions in the loss surface) and many smaller ones. To identify an effective $p$ value, we compare the effectiveness of choosing 10–50 eigenvalues in minimizing a model's sensitivity.

Table 1 summarizes our findings. We train MNIST models and measure the sensitivity by computing the Hessian trace on a trained model. We find that, when we use Top-50 of the eigenvalues, this results in the highest average accuracy of 98.92% and the lowest sensitivity (86.94%). We thus use the Top-50 eigenvalues for the rest of our paper. To stabilize the training process, we also track the trace values over the course of training and only regularize the model when the trace computed for a mini-batch is greater than the average trace values observed previously. These two strategies we employ help stabilize our training and allowing us to achieve reasonable performance and reduced model sensitivity.

Table 1: **Comparing our method using the Hessian trace from Top-$p$ eigenvalues.** Each row reports the average mean and standard deviations of the traces we compute over 1000 random samples, repeated five times across five different models.

| $p$ value | Test accuracy | Sensitivity |
|---|---|---|
| **Standard training** | $98.55 \pm 0.53$ | $128.58 \pm 61.85$ |
| **Top-1 eigenvalue** | $98.37 \pm 0.26$ | $127.55 \pm 34.51$ |
| **Top-10 eigenvalues** | $98.16 \pm 0.21$ | $126.15 \pm 63.59$ |
| **Top-25 eigenvalues** | $97.96 \pm 0.22$ | $116.10 \pm 53.77$ |
| **Top-50 eigenvalues** | $98.92 \pm 0.20$ | $86.94 \pm 38.93$ |

## 3.3 Evaluation Metrics

We introduce our evaluation metrics here to establish a clear framework for assessing our approach's effectiveness. In our evaluation of resilience against individual, single-bit corruptions Hong et al. (2019), we first define the *distribution plot*, which counts the number of bits in a model's memory representation that, when flipped, lead to the relative accuracy drop (RAD) specified on the x-axis. RAD is defined as $(A_c - A_p)/A_c$, where $A_c$ represents the classification accuracy of a model on a test set and $A_p$ denotes the accuracy of the model under parameter corruptions.

Figure 1 shows example distribution plots contrasting the two MNIST models, one trained with our training method and the other not. We flip each bit individually and record RAD. We use a 5% granularity in RAD on the x-axis for our plots. The plots show that our training method reduce the total number of bits whose flipping result in RADs and also decrease the number of bit-flips leading to a 95–100% RAD. By using this plot, we gain a deeper understanding of the severity and impact of parameter perturbations before and after the application of our training algorithm.

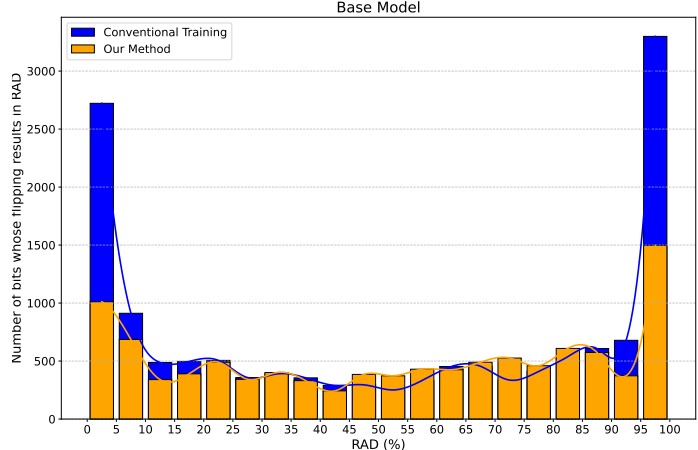

We also define an *vulnerable parameter* as one where a single-bit error results in a RAD over 10%. We use this threshold because most prior work considers a 10% RAD significant.

Figure 1: **The distribution plots** showing the number of bits in a DNN's parameters whose flipping results in specific RAD ranges on the x-axis.

Moreover, we count the number of bit-flips required to achieve near-complete accuracy depletion (e.g., RAD > 99% or an accuracy of 0.1%) to assess resilience against multi-bit corruption (Rakin et al., 2020).

## 3.4 Comparison with Prior Approaches

Next we empirically evaluate the effectiveness of our training approach compared to existing methods designed to train models with reduced sharpness. We compare against three representative methods: (1) $\ell_2$-regularization, which has been shown empirically to reduce the sharpness of a model in literature (Wu

et al., 2020); (2) AdaHessian (Yao et al., 2021b), a second-order optimizer demonstrated to be effective in reducing the sharpness; and (3) Sharpness-aware minimization (SAM) (Foret et al., 2021), a training method specifically designed to reduce the sharpness.

**Methodology.** We train MNIST and CIFAR-10 models and measure the accuracy and sensitivity. For each model, we compute the Hessian trace five times on 1000 randomly chosen training samples. For each method, we run training five times and report the average. Because in MNIST, we find that SGD struggles with optimizing our second-order objective across the hyperparameters we use, we use the RMSProp optimizer to benefit from a dynamic learning rate. We choose the base learning rate and regularization coefficient $\alpha$ from {1, 0.1, 0.01, 0.001, 0.0001}, batch size from {32, 64}, and the number of Hutchinson's steps for trace approximation from {1, 50, 100, 1000}. Through extensive hyper-parameter search, we find that using only a single step to compute the Hessian trace is the most effective.

**Results.** Table 2 summarizes our results. We show that compared to existing approaches, our method is more effective in reducing a model's sensitivity. We also employ two techniques to smooth out the Hessian regularization loss $Tr_t$ that is fluctuating over training epochs: (1) Min-max optimization: normalizing the loss based on the min and max values of the eigenvalues $\lambda_t$ as: $Tr_{t_{norm}} = Tr_t - min(\lambda_t)/max(\lambda_t) - min(\lambda_t)$, where $Tr$ denotes the Hessian trace and $t$ denotes the current step; and (2) the technique that only considers the loss when its value is greater than the one $\tau$ observed in the previous epoch (see line 14–19 in Algorithm 1). We additionally use this approach to determine and compare the impact of regularization coefficient $\alpha$, and we show that setting $\alpha$ to one for MNIST and to $\alpha$ to $10^{-2}$ for CIFAR-10 achieves the lowest sensitivity. For the rest of our experiments, we use this training configurations.

Table 2: **Comparison to existing training methods.** We compare the accuracy and the sensitivity from existing approaches to our method. The metrics are computed across five different models, and the sensitivity are computed over 1000 samples randomly chosen from the training data. We set $p$ to 50.

| Training Method | MNIST | | CIFAR10 | |
|---|---|---|---|---|
| | Acc. | Sensitivity | Acc. | Sensitivity |
| **Baseline** | 98.90 | $123.68 \pm 63.79$ | 92.43 | $3808.91 \pm 803.19$ |
| **L2-Regularization** | 97.30 | $128.23 \pm 52.42$ | 91.72 | $4117.33 \pm 1032.42$ |
| **AdaHessian** | 98.88 | $126.67 \pm 70.82$ | 92.68 | $3717.55 \pm 931.80$ |
| **SAM** | 97.15 | $134.08 \pm 75.04$ | 92.15 | $3676.89 \pm 899.82$ |
| **Ours** (Min-max) | 98.65 | $128.72 \pm 68.50$ | 92.34 | $3571.88 \pm 924.67$ |
| **Ours** ($\alpha$ to $10^{-4}$) | 98.78 | $126.67 \pm 70.82$ | 92.58 | $3543.33 \pm 952.44$ |
| **Ours** ($\alpha$ to 1) | 98.92 | $86.94 \pm 38.93$ | – | – |
| **Ours** ($\alpha$ to $10^{-2}$) | – | – | 92.71 | $2729.53 \pm 762.94$ |

Table 3: **Contrasting our approach to existing second-order training methods.** BaseNet is trained on MNIST using AdaHessian, SAM, HERO (Yang et al., 2022), and our method. Column 4 reports the number of vulnerable parameters and Column 6 their ratio to the total parameters.

| Method | Acc. | # Params | Vuln. Params | Ratio |
|---|---|---|---|---|
| **Baseline** | 98.73% | | 10,544 | 48.27% |
| **AdaHessian** | 98.88% | | 10,473 | 47.72% |
| **SAM** | 97.15% | 21,840 | 10,621 | 48.63% |
| **HERO** | 98.27% | | 10,274 | 47.04% |
| **Ours** | 98.66% | | 8,482 | 38.83% |

Moreover, we evaluate these existing training approaches in enhancing a model's resilience to bitwise corruptions. Table 3 compares the reduction in the number and ratio of vulnerable parameters for each method relative to the standard training baseline. We also include a comparison with HERO (Yang et al., 2022), a method specifically designed to train models with guarantees against bounded, small perturbations. We find that all the existing approaches reduce the number of vulnerable parameters by only a small margin (0–1.4%), whereas ours achieves a 10% reduction without any substantial accuracy drop. Interestingly, HERO, designed to enhance resilience to small perturbations, offers *no resilience* to bitwise corruptions.

## 4 Empirical Evaluation

We now evaluate our approach, focusing on three research questions: **(RQ1)** How effective is our method in reducing vulnerability? **(RQ2)** How can we characterize the vulnerability reduction achieved by models trained with our method? **(RQ3)** Can our approach synergize with existing non-training-time defenses?

### 4.1 Experimental Setup

**Datasets.** We utilize three classification benchmarks: MNIST (LeCun et al., 2010), CIFAR-10 (Krizhevsky, 2009), and ImageNet (Russakovsky et al., 2015), which are standard datasets for assessing prior defenses.

**Models.** We evaluate on four DNN architectures, covering a spectrum from simple CNNs to residual networks and Transformers. For MNIST, we use two feed-forward networks: one with two convolutional and two fully-connected layers, and LeNet (Lecun et al., 1998). For CIFAR-10 and ImageNet, we adopt architectures popular in the community: ResNets (He et al., 2016) and a Transformer, DeiT-Tiny (Touvron et al., 2021).

**Metrics.** Our metrics are described in §3.3. Against individual single-bit corruptions, we compare the number of vulnerable parameters and the total number of bits whose flipping results in a specific RAD range, e.g., 90–100%, as illustrated in the distribution plot. We also quantify the model's resilience to (targeted) bit-flip attacks by measuring the number of bit-flips required to reduce accuracy to 0.1%. Please refer to Appendix A for additional details on our experimental setup.

### 4.2 Quantifying Enhanced Model Resilience

We quantitatively analyze the resilience of models produced by our training method against bitwise parameter corruptions, focusing on the effectiveness of our approach in reducing vulnerability **(RQ1)**.

**Methodology.** We first evaluate our models against an adversary capable of inducing individual, single-bit corruptions in memory (Hong et al., 2019). This evaluation captures a model's sensitivity under worst-case bitwise corruption (i.e., the extent to which a single bit-flip can degrade accuracy). We then extend our evaluation to adversaries capable of inducing targeted, multi-bit corruptions in memory (Rakin et al., 2020). Unlike the single-bit attackers, this scenario assumes an adversary that progressively flips bits in memory until they achieve a targeted accuracy (e.g., 0.1%), effectively reducing the model's performance to near zero.

We individually flip all 32 possible bits in each model parameters for the MNIST models (BaseNet and LeNet). However, conducting the same exhaustive analysis for CIFAR-10 and ImageNet models is computationally infeasible; for example, it would take take ∼503 days to test all the bits in ResNet18 for CIFAR-10. To address this, we employ the speed-up techniques proposed by Hong et al. (2019). Because the bits most likely to cause substantial accuracy drops are the MSBs of the exponents, we restrict our analysis to exponent bits for CIFAR-10 and only the MSB of the exponents for ImageNet. For ImageNet, in ResNet50, we test a randomly chosen 50% of the parameters in all convolutional layers and all parameters in fully-connected layers. For the Transformer model, we test all MSBs of the exponent.

Table 4: **Enhanced resilience of models trained with our method.** We compare the resilience of models trained with our method (*Ours*) to those trained without (*Baseline*) against a single-bit error in their parameters. # Params are # Bits are the total number of parameters and bits examined, and *Acc.* and *Vuln.* refers to accuracy and erratic, respectively. Δ is the reduction in vulnerable parameter ratios.

| Task | DNN | # Params | # Bits | Baseline | | | Ours | | | |
|---|---|---|---|---|---|---|---|---|---|---|
| | | | | Acc. | Vuln. Params | Ratio | Acc. | Vuln. Params | Ratio | Δ |
| **MNIST** | **BaseNet** | 21,840 | 0.69M | 98.73 | 10,544 | 48.27% | 98.66 | 8,482 | 38.83% | -9.44% |
| | **LeNet** | 44,470 | 1.4M | 99.61 | 20,712 | 46.57% | 98.91 | 15,383 | 34.59% | -11.98% |
| **CIFAR-10** | **ResNet18** | 11M | 88M | 92.43 | 4.4M | 40.12% | 93.68 | 3.7M | 33.6% | -6.52% |
| **ImageNet** | **ResNet50** | 13.79M | 13.79M | 76.13 | 5.3M | 43.35% | 75.09 | 4.5M | 36.59% | -6.76% |
| | **DeiT-tiny** | 4.5M | 4.5M | 72.19 | 1.9M | 43.67% | 71.93 | 1.6M | 36.84% | -6.83% |

**Results.** Table 4 summarizes our results. We first note that our Hessian-aware training preserves model accuracy. In all cases, the Acc. columns show that, there are negligible differences in Top-1 classification accuracy between the baseline models and those trained with our method. More importantly, our approach reduces the number of vulnerable parameters by 6.5–12.0%: In MNIST models, we observe a 10% reduction, while the reduction is 6.5–6.8% in the CIFAR-10 and ImageNet models. We attribute this difference in reductions to the smaller number of vulnerable parameters in CIFAR-10 and ImageNet models, which shows 3–8% fewer vulnerable parameter ratios compared to MNIST models. Surprisingly, for the ImageNet models,

even if we employ a training strategy that fine-tunes only the last fully-connected layer, (the most sensitive layer), our method still enhances their resilience to individual, single-bit corruptions by 6.8%.

We also summarize our results against the multi-bit corruption adversary (Rakin et al., 2019) in Table 5. This attack employs a progressive bit-search that iteratively identifies the next bit to flip that maximizes performance degradation while minimizing the number of bit-flips needed. To be consistent with the results from the original study, we evaluate our method on ImageNet models and use their attack configurations. We report the number of bit-flips required to degrade model accuracy to 0.1%. The

Table 5: **Resilience of our models against multi-bit corruptions.** We report the original Top-1 accuracy of the models, along with the number of bit-flips required to reduce their accuracy to 0.1%.

| Task | Model | Acc. | # Bit-flips Needed | |
|---|---|---|---|---|
| | | | **Baseline** | **Ours** |
| **ImageNet** | **ResNet18** | 69.57 | 13 | 31 |
| | **ResNet50** | 75.33 | 11 | 29 |

table shows that models trained with our method require ∼3× more bit-flips to reach this threshold. This increase is practically significant as most real-world practical fault-injection attacks like Rowhammer are limited to flipping fewer than 10–20 bits in secure DRAM modules (Jattke et al., 2022).

## 4.3 Characterization of the Enhanced Model Resilience

We delve deeper into how various properties of a model influence its resilience to bitwise errors **(RQ2)**.

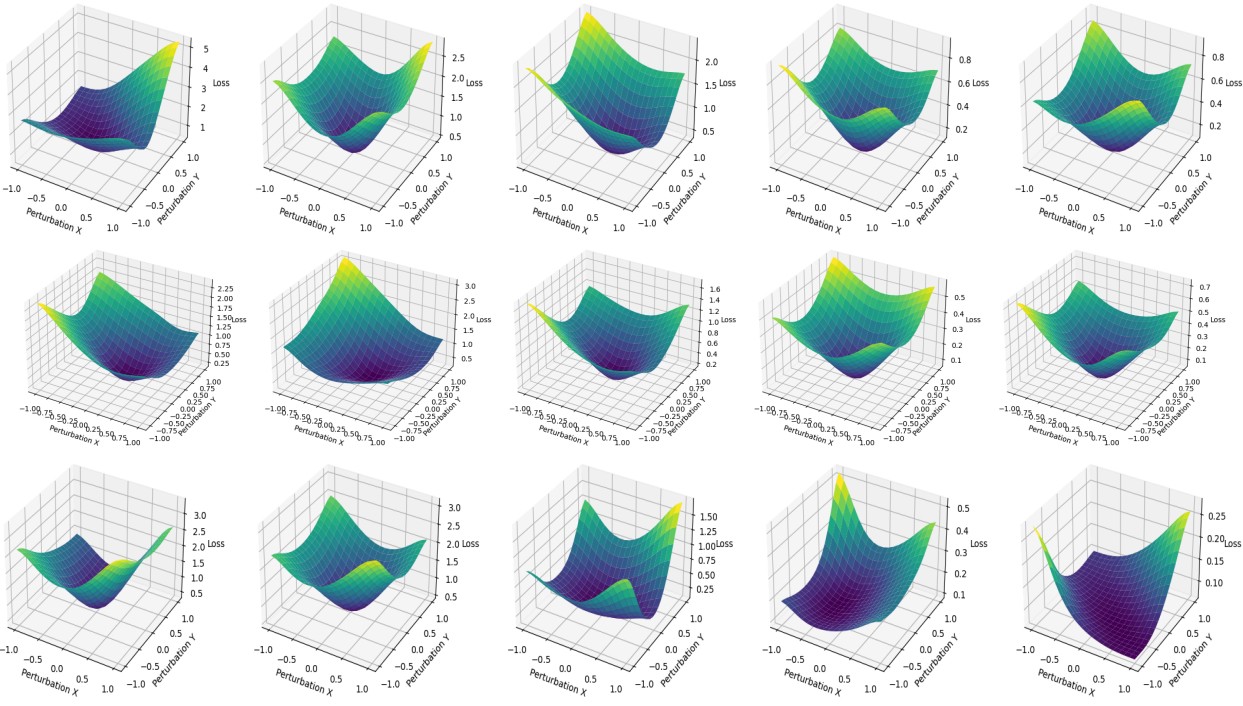

Figure 2: **Visualizing LeNet's loss landscapes.** From top to bottom, each row corresponds to standard training, HERO (Yang et al., 2022), and our method. From left to right, we visualize the first two convolutional layers followed by the three fully-conected layers.

**Visualizing the loss landscape.** We first analyze whether the models trained with our method have a *flatter* loss surface compare to the baselines. Prior work (Yang et al., 2022) has shown that DNNs with flatter loss landscapes are more resilient to parameter-level errors, since bit-flips are less likely to cause large in crease in loss when the surface lacks sharp regions. To this end, we adopt the visualization technique proposed by Li et al. (2018): We choose two random vectors with the same dimensionality as the model parameters and incrementally perturb the parameters along each direction while measuring the loss of the

perturbed model. Figure 2 shows the resulting loss landscape for each layer of the LeNet models trained on MNIST. From left to right, we visualize the five layers in order from the input.

We show that our method effectively increases flatness across all layers, with the notable improvements in the layers close to the output. In the last three columns of Figure 2, which correspond to the fully-connected layers, our method (bottom row) produces substantially flatter loss surfaces compared to the baselines (top two rows). We also observe that our approach is less effective at increasing the flatness of the layers close to the input (the first two convolutional layers). We find that this trend holds for CIFAR-10 and ImageNet models (see Appendix C). This, however, is not a weakness of our method, as layers closer to the output contain more parameters than earlier ones and are more prone to causing drastic changes in loss values when subjected to perturbations. While not intended, this further enhances our method's effectiveness.

Table 6: **Comparing the effectiveness of our approach in convolutional (Conv.) and fully-connected (FC) layers.** Ours refers to models trained with our approach, while Baseline is models trained without. Column 4 reports the # of parameters in Conv. or FC layers, with the parenthesis indicating their ratio within each model. All other values show vulnerable parameters and their ratios. The last two columns show the $\Delta$ in these metrics. For ResNet50 Conv. layer, $\dagger$ refers to results from 50% sampled parameters.

| Task | Model | Layers | # Params | Baseline | | Ours | | $\Delta$ | |
|---|---|---|---|---|---|---|---|---|---|
| | | | | Vuln. Params | Ratio | Vuln. Params | Ratio | Vuln. Params | Ratio |
| **MNIST** | **BaseNet** | **Conv.** | 5,280 (24.2%) | 3,003 | 56.87% | 2,695 | 51.04% | -308 | -5.83% |
| | | **FC** | 16,560 (75.8%) | 7,544 | 45.55% | 5,811 | 35.09% | -1,733 | -10.46% |
| | **LeNet** | **Conv.** | 2,616 (5.9%) | 1,719 | 65.71% | 1,475 | 56.38% | -244 | -9.33% |
| | | **FC** | 41,854 (94.1%) | 20,013 | 47.81% | 14,903 | 35.61% | -5,110 | -12.20% |
| **CIFAR-10** | **ResNet18** | **Conv.** | 11.2M (99.7%) | 4.4M | 40.07% | 3.7M | 33.57% | -0.7M | -6.50% |
| | | **FC** | 5,120 (0.03%) | 2,297 | 44.86% | 1,321 | 25.80% | -976 | -19.06% |
| **ImageNet** | **ResNet50** | **Conv.** | $\dagger$23.5M (53.5%) | 4,516,162 | 38.23% | 3,802,648 | 32.19% | -713,514 | -6.04% |
| | | **FC** | 2.04M (46.5%) | 766,940 | 37.59% | 656,493 | 32.18% | -110,447 | -5.40% |

**Enhanced resilience in convolutional vs. fully-connected layers.** Our previous analysis of the loss surfaces suggests that the method tends to reduce the sensitivity (i.e., sharpness) of the later layers. Since most feed-forward neural networks consists of convolutional layers followed by fully connected layers for classification, we analyze whether the resilience has indeed increased in the fully connected layers.

Table 6 summarizes our findings. In most cases, we observe that the reduction in the ratio of vulnerable parameters in fully connected layers is 2.4–13.4% greater than that in convolutional layers. Particularly, for the ResNet18 trained on CIFAR-10, our Hessian-aware training reduces the vulnerable parameter ratio by 19.1%. This result implies that network architectures with many fully connected layers, such as BaseNet or LeNet, can benefit more from our method. However, in the ResNet50 model trained on ImageNet, we observe the same ratio of vulnerable parameters in both Conv. and FC layers. We attribute this to the use of our speed-up technique: for ImageNet we flip only the 31st bit position in each parameter. Parameters in Conv. layers are solely vulnerable to flipping 31st bit, whereas those in FC layers can also cause accuracy drops when other exponent bits are flipped (see Appendix F for our empirical analysis of this behavior).

**Reduction in the number of bits causing RAD within specific ranges.** To gain a deeper insight into the enhanced resilience by our approach, we contrast the distribution plots between two models: one trained with our method and the other without. Figure 3 presents the plots for the LeNet (left) and ResNet18 (right) models. Please refer to Appendix B for additional plots of the ImageNet models. In both models, our approach significantly reduces the number of bits in two regions: (1) bits whose flipping causes substantial performance loss (90–100% RAD) and (2) bits whose flipping causes only minor accuracy drops (0–10% RAD). These results imply that our training method reduces the likelihood of a model's performance degrading to a random output generator by 25–50% due to a single bit-flip.

## 4.4 Synergy with Existing (System-level) Defenses

We further examines the synergy between system-level defenses and models trained with our method **(RQ3)**.

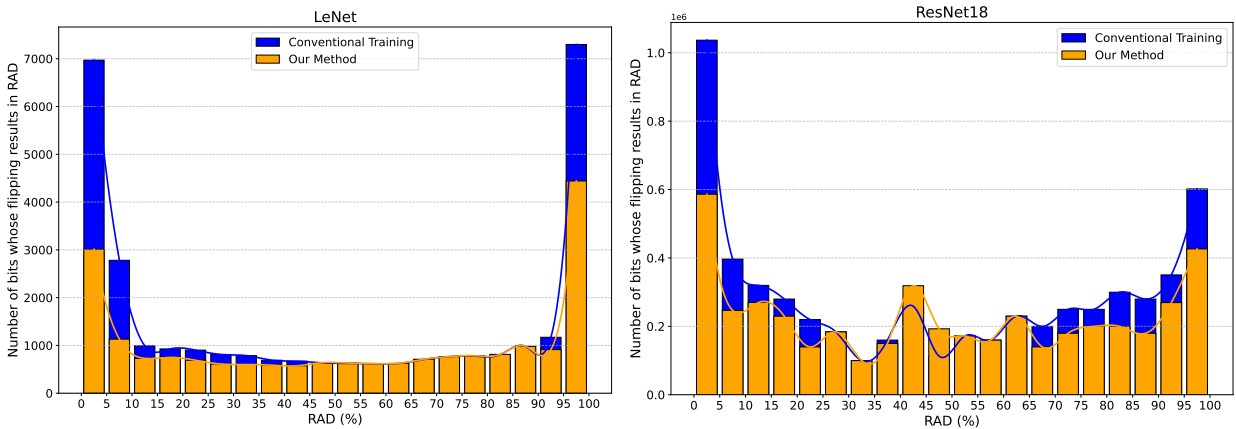

Figure 3: **The distribution plots** computed on LeNet in MNIST (left) and ResNet18 on CIFAR10 (right).

**NeuroPot** (Liu et al., 2023) injects honey (or decoy) neurons into a model at locations likely to be targeted by an adversary, without causing any significant accuracy drop. Once these honey neurons are injected, the defense needs system-level supports, such as an additional checksum modules or a secure memory area like trusted execution environments (TEEs), to store the original parameters for detection and recovery. Because our method reduces the number of vulnerable parameters, NeuroPot could benefit by requiring fewer honey neurons. Table 7 summarizes the benefits of combining NeuroPot with models trained with our method. We test LeNet for MNIST, ResNet18 for CIFAR-10, and ResNet50 for ImageNet.

NeuroPot, when combined with our method, enhances the resilience or reduces space complexity. If the number of honey neurons ($\# h_n$) is fixed, models trained with our method require twice as many bit-flips ($\# bits$) to cause a 10% accuracy drop compared to the baselines. If an adversary is allowed to cause a 10% accuracy drop with the same $\# bits$, our models need 60–80% fewer honey neurons, thereby improving inference time and storage efficiency. For instance, reducing to 30 $h_n$ in the ResNet50 ImageNet model decreases inference time by 55.9% (0.37s vs. 0.84s) and storage overhead by 65% (34KB vs. 99KB).

Table 7: **Synergy with NeuroPot.** We report the number of honey neurons ($\# h_n$) and the number of bit-flips ($\# bits$) required to cause an accuracy drop over 10%, averaged over 5 runs.

| Model | Baseline | | | Ours | | | | |
|---|---|---|---|---|---|---|---|---|
| | **Acc.** | $\# h_n$ | $\#$ bits | **Acc.** | $\# h_n$ | $\#$ bits | $\# h_n$ | $\#$ bits |
| **LeNet** | 99.32 | 25 | $11 \pm 2$ | 99.47 | 25 | $21 \pm 4$ | 10 | $12 \pm 2$ |
| **ResNet18** | 92.17 | 50 | $15 \pm 3$ | 92.13 | 50 | $34 \pm 3$ | 20 | $15 \pm 4$ |
| **ResNet50** | 76.09 | 150 | $17 \pm 3$ | 76.01 | 150 | $31 \pm 5$ | 30 | $16 \pm 4$ |

**RADAR** (Li et al., 2021) is a checksum-based defense, which stores *golden signature* for a group of weights and compares this signature at runtime with the current model signature. We adapt this scheme to store the golden signature of vulnerable parameters, further enhancing resilience at runtime. RADAR involves a trade-off: larger group sizes increase the likelihood of checksum failure (since multiple bits within a group may flip), whereas smaller group sizes provide stronger protection but increase time and space complexity to hold more checksum values. In our evaluation, we test two models: ResNet20 for CIFAR-10 and ResNet18 for ImageNet. We use the group sizes used in the original study—8 for ResNet20 and 512 for ResNet18.

Table 8: **Synergy with RADAR.** We report the accuracy after 15 random bit-flips (Acc.) and the accuracy after the recovery using RADAR (Recovery).

| Model | Initial Acc. | Baseline | | Ours | |
|---|---|---|---|---|---|
| | | **Acc.** | **Recovery** | **Acc.** | **Recovery** |
| **ResNet20** | 90.13 | 18.01 | 81.13 | 27.93 | 88.23 |
| **ResNet18** | 69.34 | 0.19 | 60.18 | 15.29 | 64.88 |

We demonstrate the synergy of our approach with RADAR in two ways. (1) We evaluate the accuracy of our models after RADAR recovers parameters whose checksums fail. As shown in Table 8, with RADAR, models trained with our method achieve higher accuracy recovery–88.23% for ResNet20 and 64.88% for ResNet18–compared to baseline models, which recover to 81.13% and 60.18%, respectively.

Table 9: **Impact of our models on RADAR's time and space complexity.** G denotes the group size and Acc. Rec. refers to the accuracy recovery achieved using RADAR. Inference time is measured in milliseconds (ms), and space is measured in kilobytes (kB).

| Model | Baseline | | | | Ours | | | |
|---|---|---|---|---|---|---|---|---|
| | G | Acc. Rec. | Time | Space | G | Acc. Rec. | Time | Space |
| **ResNet20** | 8 | 81.13 | 0.06ms | 8.2kB | 64 | 80.93 | 0.02ms | 3.2kB |
| **ResNet18** | 512 | 60.18 | 3.32ms | 5.6kB | 1024 | 59.47 | 1.86ms | 2.95kB |

(2) We show that our models enable a more favorable trade-off between protection and time/space complexity. Table 9 reports our results. We progressively increase the group sizes in our models until their recovery accuracy matches that of the baseline models. For example, in ResNet20 trained on CIFAR-10, increasing the group size by 4× (to 64) results in a reduced accuracy recovery of 80.93%, which is still comparable to the baseline model's runtime recovery (81.13%). The choice, however, offers substantial efficiency gains: a 69% reduction in both runtime and space complexity compared to the baseline CIFAR-10 model.

## 5 Discussion

**Increase in computational demands at training.** We evaluate the overhead of our training method in terms of actual training wall-time measured in PyTorch on a NVIDIA Tesla V100 GPU. In Appendix E we present our results. The Hessian-aware training incurs overhead that scales with the size of the model; a 4–6× times increase in computations for MNIST models, and a 10× times increase in overhead for CIFAR-10 models. Existing works utilizing second-order properties during training take a completely different approach compute the Hessian and its eigenvalues: they employ weight perturbations (Foret et al., 2021) or only the trace approximation (Yao et al., 2021b) to minimize sharpness of the loss landscape. The increase in computation in our approach is primarily attributed to the large Hessian and eigenvalues we need to compute with respect to model parameters, which is not optimized for popular deep-learning frameworks. To reduce the computational overhead during training, we employ a layer sampling technique. As prior work identifies the last layers to be most susceptible to bitwise errors (Hong et al., 2019), we believe only computing Hessian trace on last few layers can aid resilient model training. Our results for large-scale models, such as ResNet50 in ImageNet, show that this technique significantly reduces the computational overhead from 10× times to 1.18× times, being equally effective in enhancing model resilience. We also note that this overhead is incurred only once during training, while inference time remains unchanged.

**Hardware-level defenses.** Many hardware-level defenses are designed to mitigate RowHammer (Kim et al., 2014), a software-induced attack that causes a targeted DRAM row to leak capacitance by repeatedly accessing its neighboring rows. Kim et al. (2014) have proposed a defense that proactively refreshes rows that are frequently accessed, as they are at higher risk of being targeted by the attack. Panopticon (Bennett et al., 2021) leverages a similar idea: it employs hardware counters for each data row in DRAM and refreshes the rows when the counter reaches a predefined threshold. Instead of refreshing the rows at high risks, Saileshwar et al. (2022) propose swapping them with safe memory regions. Di Dio et al. (2023) use the error correction codes as a mechanism for triggering such swapping. DRAM-Locker (Zhou et al., 2023b) leverages a lock-table in SRAM. If the addresses of the high-risk rows are stored in the lock-table, any access this addresses without the unlock command will be denied. These defenses mainly protect data rows at high risk of being targeted. Our work reduces the number of data rows in a model whose perturbations lead to significant accuracy loss, and therefore, potentially decreasing their runtime and space overheads.

## 6 Conclusion

Our work presents a novel training algorithm designed to reduce a model's sensitivity to parameter variations, thereby enhancing its resilience to bitwise corruptions in model parameters. We focus on the model's second-order property, the Hessian trace, and develop an objective function to directly minimize it during training. We extensively compare our approach with existing methods for improving model resilience and demonstrate its effectiveness. We evaluate our approach by testing a model's performance under both single-bit and multi-bit parameter corruptions. Our method reduces the number of erratic parameters by 6–12%, and decreases those causing a 90–100% RAD drop by 20–50%. We also increase the number of bit-flips required

by a multi-bit adversary to induce substantial accuracy drops. Moreover, we demonstrate the synergy when combined with system-level defenses to protecting models against parameter-corruption attacks. We hope our work will inspire future work on the safe deployment of DNNs in emerging computing platforms.

**Broader Impact Statement**

This paper presents a hessian-based DNN training approach to enhancing the inherent resilience of a DNN model to unbounded bitwise errors frequently observed in emerging hardware devices. By exploring Hessian-based training and evaluating their efficacy across various datasets and architectures, this work significantly advances understanding and mitigation of hardware-induced vulnerabilities in DNNs. The proposed methodology not only improves the resilience of the models but also bridges critical gaps in deploying machine learning systems in error-prone devices. This research has potential implications for industries relying on error-prone hardware where resilience to hardware perturbations is paramount. By ensuring reliable performance under challenging conditions, our work contributes to the broader adoption of DNNs across diverse fields, fostering innovation and reliability in next-generation computing systems.

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

# A  Detailed Experimental Setup

We describe our experimental setup in detail. All experiments use Python v3.11.4[1] with Pytorch v2.1.0[2] and CUDA v12.1[3] for GPU acceleration. We run our experiments on two systems: (1) a node with a 48-core Intel Xeon Processor, 768GB of memory, and 8 NVIDIA A40 GPUs. (2) a node with a 56-core Intel Xeon Processor, and 8 Nvidia Tesla H100 GPUs. We achieve a substantial speed-up in running our evaluation script by utilizing the parameter-level parallelism on the two systems.

We use the following hyper-parameters to train/fine-tune our models.

**MNIST.** We use a network architecture (Base) and LeNet in prior work (Hong et al., 2019). For regular training, we used an SGD optimizer with a learning rate of 0.1 (adjusting by 0.25 every 10 epochs), batch size of 64, and 0.8 momentum. We train our models for 40 epochs. To train the same network using our Hessian-aware training, we used $\lambda$ (line 16 of algorithm 1) value of 1 as per our findings in table 2. We use the RMSProp optimizer, keeping all the other hyper-parameters the same as the regular training.

**CIFAR-10.** We use ResNet18. For the standard training, we use SGD, 0.02 learning rate, 32 batch-size, 0.9 momentum. We train our models for 90 epochs. We adjust the learning rate by 0.5 every 15 epochs. We use the RMSProp optimizer and $\lambda$ value of $10^{-2}$ to train the same model with our approach.

**ImageNet.** We take the ResNet50 architecture pretrained on ImageNet (available at Torchvision library[4]). Instead of retraining the ResNet50 from scratch, we fine-tune the model on the same ImageNet dataset.

During fine-tuning, computing the Hessian matrix has a high computational demand. We thus leverage our observation in §4.3 and focus on the layers closer to the model output. We only compute Hessian eigenvalues and trace on the last layer and fine-tune the entire model using our method. The hyper-parameters have been kept as Torchvision's original training hyper-parameters [5]), but using the RMSProp optimizer.

For fine-tuning the Diet-tiny ViT model on ImageNet, we use the same technique for hessian and eigenvalue computation. We take the pre-trained model from HuggingFace (available at [6]) and fine-tune it. We adopt the original training setup from (Touvron et al., 2021), that uses batch size of 32, learning rate 0.1 and reducing by 0.1 every 30 epoch, momentum of 0.9, weight decay $10^{-4}$ and 90 epochs training cycle except we use RMSProp. We experimentally found $\lambda$ value of $10^{-3}$ to achieve better generalization in ImageNet.

# B  Distribution Plot Computed on ImageNet Models

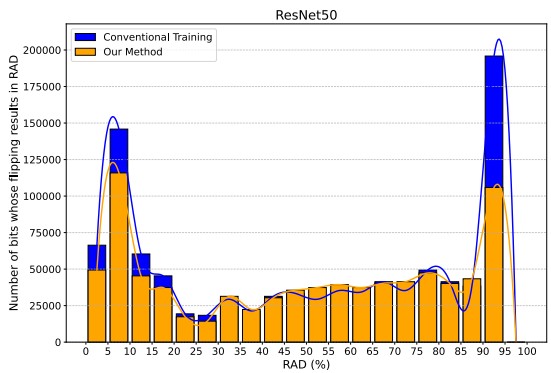

Figure 4: **The distribution plot for ResNet50.**

We show the distribution plot computed on the ImageNet models in Figure 4. We observe that fine-tuning the pre-trained ResNet50 achieves an enhanced resilience to bitwise errors in parameters. It reduces the number of corruptions leading to an accuracy drop in the range between 0-30%. We also reduce the number of parameters whose bitwise error leads to an accuracy drop of over 90% by half. Our result on ImageNet is particularly interesting because, even if we do not train our model with the Hessian trace computed on the entire layers, we can offer enhanced resilience to a DNN model. While in MNIST and CIFAR-10 models, we see the number of parameters causing accuracy loss of 0–5%, in our fine-tuned ImageNet model, we find a greater number of parameters causing accuracy drops at 5–10% bin.

---

[1]Python: https://www.python.org

[2]PyTorch: https://pytorch.org/

[3]CUDA: https://developer.nvidia.com/cuda-downloads

[4]Pre-trained PyTorch models: https://pytorch.org/vision/stable/models.html

[5]https://github.com/pytorch/vision/tree/main/references/classification

[6]DeiT-tiny: https://huggingface.co/facebook/deit-tiny-patch16-224

## C   Visualizing Loss Landscapes of Layers with Residual Connections

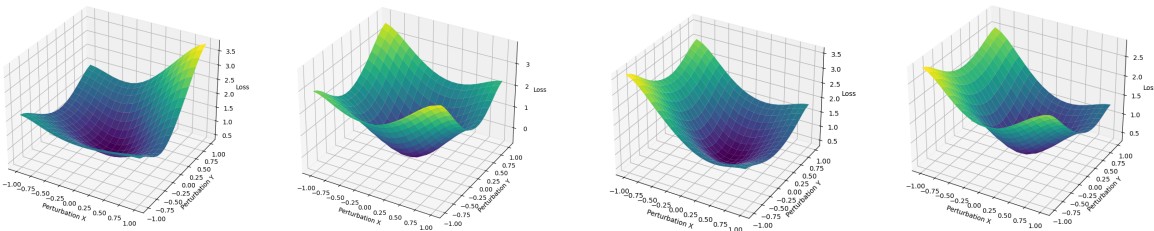

Figure 5: **Comparing loss landscapes of the convolutional layers within a residual block.** The left two are from the regularly-trained models, and the right ones are from those trained with our method.

Prior work (Li et al., 2018) has visually shown that convolutional layers with residual connections tend to have flatter loss surfaces. In such layers, we hypothesize that our approach is less effective in reducing the sensitivity. Figure 5 shows the loss landscapes from two convolutional layers in ResNet18 models trained on CIFAR-10. We observe that the loss landscapes visually look similar to each other, implying that our approach was less effective in reducing the Hessian trace of these layers. This does not mean that these layers are particularly susceptible to bitwise errors in parameters. On the other hands, these convolutional layers already have some resilience to bitwise errors in parameters.

## D   Numerical Perturbations Causing Accuracy Drop over 10%

We analyze how resilient a model becomes to actual parameter value changes caused by single bitwise errors. Using the parameter values before any corruption and after causing a single-bit error, we compute the changes in the numerical values on two models (one regularly-trained, and the other trained with our approach). Figure 6 shows our results from the Base, LeNet and ResNet18 models. We demonstrate that DNN models trained with our method requires a greater numerical variations to cause a RAD drop over 10% than those trained using regular training methods. Based on our observation that most single-bit errors cause a bit-flip in the most significant bit of the exponent (i.e., the 31st-bit), the numerical variations required to cause a large performance loss go beyond the range that floating-point representation in modern systems can hold.

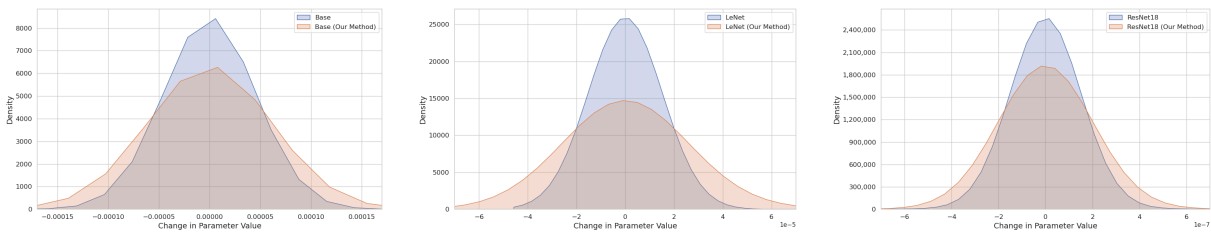

Figure 6: **Comparison of numerical perturbations required to cause an accuracy drop over 10%.** From left to right, the results correspond to Base and LeNet in MNIST, and ResNet18 in CIFAR-10.

## E   Overhead of Hessian Aware Training

In this section, we analyze the computational complexity of our training method. Let $P$ denote the number of model parameters, $L$ the number of layers, $p$ the number of Hutchinson probe vectors, and FLOPs$_l$ the number of floating-point operations required for layer $l$.

A standard forward-backward pass has computational complexity:

$$F_{SGD} = \Theta \left( \sum_{l=1}^{L} \text{FLOPs}_l \right).$$

A Hessian-vector product (HVP) has a cost comparable to an additional forward-backward pass, i.e.,

$$F_{HVP} = \Theta \left( \sum_{l=1}^{L} \text{FLOPs}_l \right).$$

Since our method performs one standard gradient update and $p$ HVPs per iteration, the total cost per training step becomes:

$$F_{step} = \Theta \left( F_{SGD} + p * F_{HVP} \right) = \Theta \left( (1+p) \sum_{l=1}^{L} \text{FLOPs}_l \right).$$

Because the total number of FLOPs scales approximately linearly with the number of parameters $P$, this simplifies to:

$$F_{step} = \Theta((1+p)P).$$

Our layer-sampling strategy further restricts the Hessian computations to the final layer $l'$. In this case, the cost reduces to:

$$\sum_{l=1}^{L} \text{FLOPs}_l \rightarrow \sum_{l=1}^{L} \text{FLOPs}_{l'}$$

which significantly lowers the practical overhead. This aligns with our empirical findings in Table 10, where the overhead is reduced to approximately $1.18\times$ for ImageNet-scale models.

We measure the overhead we used the optimal hyper-parameter setup described in Appendix A for all our models. We run training for 5 times, and report the per epoch training time. The measured values are presented in Table 10. Result shows that hessian aware training has a 4x-6x overhead for MNIST models. For the ResNet18 model trained on CIFAR-10, the overhead increases further due to ResNet18's larger architecture and higher number of parameters compared to smaller MNIST models. We employ layer sampling technique to reduce this overhead in our larger models. We calculate the Hessian eigenvalues and trace on the last layers and fine tune the model. Prior research (Hong et al., 2019) suggests that these final layers are the most susceptible against parameter corruption, making this a viable strategy for applying our method to large-scale models. Our result shows that adopting this method has only 1.18x computational overhead.

Table 10: **Comparing the training time of our method to baseline training in terms of runtime in PyTorch.** We report the per-epoch runtime (in seconds) for all our models trained across 3 datasets.

| Model | Dataset | Training Time | |
|---|---|---|---|
| | | Baseline | Our Method |
| **Base** | MNIST | $0.335 \pm 0.002$ | $1.362 \pm 0.0085$ |
| **LeNet** | | $0.432 \pm 0.003$ | $2.857 \pm 0.0073$ |
| **ResNet18** | CIFAR10 | $36.244 \pm 0.607$ | $341.58 \pm 9.81$ |
| **ResNet50** | ImageNet | $7275.6 \pm 18.41$ | $8647.2 \pm 25.43$ |

We conduct additional experiment on the layer-sampling technique for larger architecures like ResNet18 and ResNet50. Following the same overhaed measurement approach, We applied Hessian regularization incrementally to the layers closer to output. We start with only the last layer and extend it to the last 2, 3, and finally 4 layers of the model and compared the runtime with baseline training. Our results are presented in Table 11.

Table 11: **Comparing the training time of layer-sampling and baseline training in PyTorch.** We report the per-epoch runtime (in seconds).

| Model | Dataset | Training Time (in seconds) | | | | |
|---|---|---|---|---|---|---|
| | | **Baseline** | **L1** | **L2** | **L3** | **L4** |
| **ResNet18** | CIFAR10 | $36.244 \pm 0.607$ | $37.77 \pm 0.39$ | $43.24 \pm 0.28$ | $57.63 \pm 0.44$ | $78.24 \pm 1.13$ |
| **ResNet50** | ImageNet | $7275.6 \pm 18.41$ | $8647.2 \pm 25.43$ | $10134.7 \pm 30.21$ | $13289.5 \pm 35.76$ | $16547.8 \pm 42.15$ |

Results in Table 11 demonstrate that training overhead increases as we increase the "layers involved in Hessian eigenvalue and trace calculation." However, using only the last 1 layer of the model, we can reduce the overhead to almost the same as baseline training, making our method efficient for very large models. We note that the increased computational time is not solely due to adopting our training method. The additional time is primarily attributed to the computation of the large hessian trace and eigenvalues, which is not fully optimized for use with popular deep learning frameworks such as PyTorch. We leave further optimization of our approach as future work.

## F   Analysis of Corrupted Bit Position

The IEEE 754 standard defines the representation of floating-point numbers in modern computer systems. In this format, a 32-bit number is represented with three fields: the 1-bit sign, the 8-bit exponent, and the 23-bit mantissa. Similar to the prior work (Hong et al., 2019; Rakin et al., 2019; Yao et al., 2020a), we analyze the location of bitwise corporations that lead to an accuracy drop over 10%. Figure 7 shows our analysis results. We use a logarithmic scale in the y-axis.

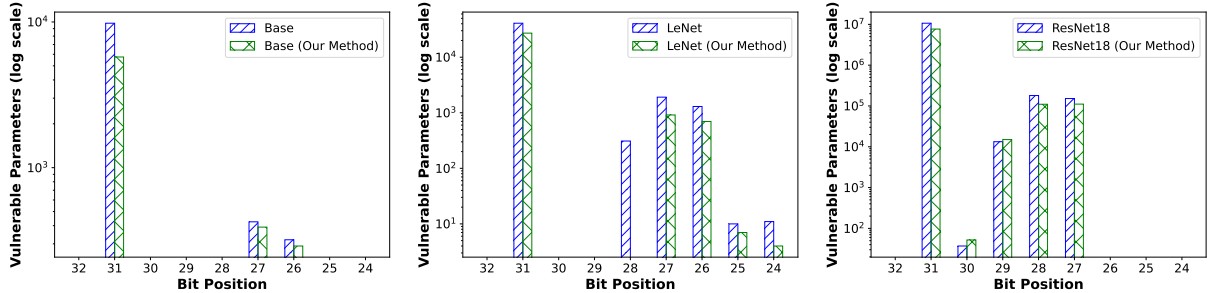

Figure 7: **Comparison of the corrupted bit positions.** From left to right, we show the analysis result from Base (MNIST), LeNet (MNIST), and ResNet18 (CIFAR-10). We only examine the sign bit and the exponent bits, as they change the numerical value of a parameter the most.

In all the models, corruption of the $31^{st}$ bit mostly leads to an accuracy drop over 10%. These corruptions account for $\sim$93% and $\sim$91.43% in the Base and LeNel models, respectively. We also observe a few bits in the $26^{th}$ and $27^{th}$ position for both Base and LeNet models and a small number of bits in the $28^{th}$ location for the LeNet model. A consistent trend is observed in the ResNet18 models in CIFAR-10, with the $31^{st}$ bit being identified as the most susceptible bit location. However, in ResNet18, we identify a few bits positioned at the $30^{th}$ and $29^{th}$ location in the exponent. In contrast to our observations from LeNet and ResNet18, there are no susceptible corruptions in the $30^{th}, 29^{th}, 28^{th}, 25^{th}$ and $24^{th}$ bit positions in the Base model.

## G   Enhanced Model Resilience to Compression

We examine the additional benefits of our approach beyond parameter resilience to bitwise errors. We are particularly interested in testing whether models trained with our method can achieve improved performance under pruning (Han et al., 2015) or quantization (Fiesler et al., 1990). These techniques reduce the size of neural networks through parameter reduction or compression, introducing optimal parameter perturbations (LeCun

et al., 1989). Although it is not the focus of our work, we study the effectiveness of our method in increasing the resilience of DNN models against these perturbations.

**Pruning.** In our evaluation, we employ global unstructured pruning (Liu et al., 2017), which operates at the individual weight level. This technique first computes an importance score for each weight and removes those with the lowest scores. We apply this pruning technique with different sparsity levels ranging from 0–100%. Figure 8 shows our pruning results on the Base and LeNet model on MNIST and ResNet18 models trained on CIFAR-10. We demonstrate that DNN models trained with our method retain accuracy better than those trained using regular training methods. Both MNIST models retains their original accuracy up to 65% parameters pruned. Beyond this point, as sparsity increases, we observe a steep decrease in accuracy. The Base and LeNet models trained using our method shows better accuracy than the regularly-trained models. Our approach surprisingly maintains accuracy further for ResNet18 model on CIFAR-10, up to 70% pruning indicating enhanced parameter-level resilience to bitwise errors. At the same sparsity level, the model trained with the conventional approach completely loses accuracy (i.e., the accuracy dropping to ∼0%).

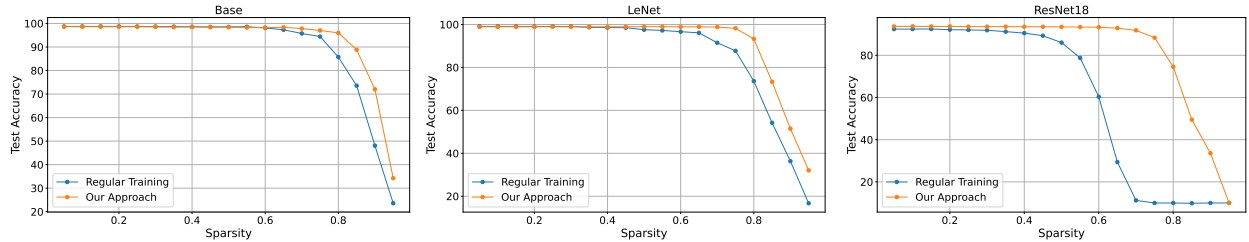

Figure 8: **Comparison of model performance under various pruning ratios.** The left and middle figure (Base and LeNet Model) is computed on the MNIST dataset, while the right one is from the ResNet18 CIFAR-10 models.

**Quantization.** Table 12 summarizes our quantization results for 8-, 4-, and 2-bit quantization of the regularly-trained models and Hessian-aware trained models. We employ layer-wise, symmetric quantization, which is the default in most deep learning frameworks. Overall, the models trained with our approach achieve better test accuracy than the regularly trained models, an additional benefit that hessian-aware training offers. Up to 4-bit quantization, both models retain the performance of their floating-point counterparts. However, when we use 2-bit precision, the accuracy of all models decreases significantly. Our models under 2-bit precision consistently achieve 1.5–14% better accuracy, indicating that these models have increased resilience to parameter value variations. Based on our observation that fully-connected layers are less sensitive than convolutional layers (see the above analysis), we employ mixed-precision quantization with 2-bit precision in fully-connected layers and 4-bit precision in convolutional layers. We demonstrate that our models achieve an accuracy of 68.8–78.7%, while the regularly-trained models achieve 48.9–68.2% model accuracy.

Table 12: **Comparison of model performance under various quantization ratios.** We compare the test accuracy of models after quantizing them with different bit-widths.

| Dataset | Model | Acc. | | | |
|---|---|---|---|---|---|
| | | **8-bit** | **4-bit** | **2-bit** | **Mixed** |
| **MNIST** | **Base** | 98.57 | 98.38 | 24.49 | 48.90 |
| | **Base (Ours)** | 98.73 | 98.70 | 38.72 | 68.84 |
| | **LeNet** | 99.10 | 98.70 | 11.85 | 57.03 |
| | **LeNet (Ours)** | 98.90 | 97.37 | 24.78 | 73.90 |
| **CIFAR-10** | **ResNet18** | 92.53 | 88.01 | 9.96 | 68.19 |
| | **ResNet18 (Ours)** | 92.36 | 90.26 | 10.28 | 78.69 |

