# OpenReview forum: "Hessian-aware Training for Enhancing DNN Resilience to Bitwise Corruptions in Their Parameters"
_TMLR — Accepted by TMLR_

### Review · Reviewer_87ho · 2025-10-14

**Summary Of Contributions:**

The paper proposes a low-bit training framework, where the trace of Hessian matrix is maximized and the trace of Hessian matrix is regarded as a regularization term. Moreover, whether to update parameters under the guidance of Hessian matrix is decided by the medium value of all the eigenvalues of Hessian matrix. The paper is well written and Algorithm 1 illustrates the pipeline clearly. Experiments on MNIST, CIFAR-10, and ImageNet are conducted.

**Audience:**

Yes

**Audience Explanation:**

Low-bit training is an emergent problem in the community of Machine Learning, not confined to Computer Vision. Large Language Models (Text, Image, Video) also expect effective low-bit training algorithms.

**Claims And Evidence:**

No

**Claims Explanation:**

Why can "updating the parameters alongside the sharpest direction (i.e., Maximizing the trace of Hessian matrix in the method)" can alleviate/address the performance gap introduced by low-bit quantization? The question is not answered in the paper, c.f., "Requested Changes".

**Requested Changes:**

Strengths:
1. The paper is well written. Training low-bit from the landscape of Neural Networks is novel.

Weaknesses:
1. About the motivation, training low-bit models alongside the sharpest direction is not convincing. What is the actual relation between "updating the parameters alongside the sharpest direction" and "quantization error". The relation is not elaborated, making the paper incomplete. Further analysis of quantization error is expected. Then, authors can build the bridge between the landscape and the quantization error.

2. The contradiction of claims. In Abstract, authors claim "learn flatter loss surfaces". However, in Algorithm 1, stochastic gradient **ascent** is utilized and the trace of Hessian is somehow increased in the iteration of training. Why can "increasing the trace of Hessian" learn "flatter" landscape. From my perspective, updating parameters alongside the large eigenvalues and increasing the trace of Hessian walks into the **sharp** point in the landscape. Authors should illustrate this point rigorously.

---

### Review · Reviewer_7ce3 · 2025-10-24

**Summary Of Contributions:**

The paper introduces a new training method for increasing the resilience of trained neural networks to hardware attacks. The basic idea is that smoothing the loss function so that small changes in the representation have little impact on the computation within neural networks.
The paper also outlines the challenges faced in developing this method.

**Additional Comments:**

As said, to me as an uninformed reader, the paper seems solid. I am not able to comment on the positioning to the existing methods. I do, very much, appreciate the insights that the authors present beyond just validating the main claims.

A few questions did come up to my mind while reading.

First, I would advise including more intuition/explanation on why the smoothness of the Hessian should reduce the change of flipping. As someone to whom neural networks are not a part of daily life, it took me a while to understand your point.

I was also wondering to what extent the architecture matters? You demonstrate your technique on CNN and fully-connected layers, but do you expect your proposal to be equally effective on other architectures?

**Audience:**

Yes

**Audience Explanation:**

The work is very relevant for practical deployment of neural networks where attacks can happen

**Broader Impact Concerns:**

the work actually helps to reduce the negative impact of attacks on machine learning models

**Claims And Evidence:**

Yes

**Claims Explanation:**

The paper is far from my field of expertise, so this is the only part I can confidently evaluate.

The main claim of the paper is that regularising the Hessian of the loss function surface, such that it is smoother, reduces the chances that flipping a bit in the numeric representation of numbers changes the output of a neural network.

The experiments clearly demonstrate that the number of vulnerable parameters is reduced by smoothing out the Hessian.
moreover, I appreciate the empirical demonstration for the design choices such as the number of eigenvalues to consider.

**Requested Changes:**

I quite like this paper -- it presents a simple idea, it properly evaluates it in my opinion, and offers additional insights obtained during development. I am not at all knowledgeable on the topic, but I have learned something from this paper.

The only aspect that I felt the paper is lacking is in addressing the related work. right now, this discussion is split between the Introduction and Discussion. I would advise having a separate section which clearly points out the differences to other methods.

Additionally, you demonstrate that setting the $alpha$ parameter correctly is important -- only for certain values does your approach significantly outperforms the competition. given the increased training costs of your method, how easy is it to choose that parameter correctly? This is an important issue you should address.

---

### Review · Reviewer_YRhN · 2025-12-01

**Summary Of Contributions:**

The authors of the paper propose a Hessian-aware training procedure that directly regularizes the (approximate) Hessian trace of the loss with respect to network parameters, with the goal of making models less sensitive to bit-flip corruptions in their stored weights. The authors adapt Hutchinson-style trace estimation and evaluate on the proposed algorithm on several models and datasets: MNIST, CIFAR-10 and ImageNet with CNNs and a Vision Transformer. Authors demonstrate that the proposed approach improved resilience of models to parameter corruptions. Authors additionally analyse what factors that may influence this increased resilience of the models towards the bit-flip corruptions.

**Audience:**

Yes

**Audience Explanation:**

The work sits at the intersection of robustness, optimization and systems and is interesting to researchers in these areas, for example for the researchers in robust and reliable ML, who study failures induced by hardware faults.

**Claims And Evidence:**

No

**Claims Explanation:**

The paper is clearly written, easy to follow, and the core problem is well motivated: designing training procedures that make neural networks more resilient to weight bit flips. The experimental section is reasonable in terms of datasets and architectures. However, from my perspective there are several critical weaknesses in the current framing and evaluation, so I do not think the key claims are yet supported by sufficiently clear and convincing evidence.

1. **Missing connection to Lipschitz regularized models**

   There is a substantial literature on directly enforcing Lipschitz continuity of neural networks to improve robustness and stability, for example Gouk et al., “Regularisation of neural networks by enforcing Lipschitz continuity” (Machine Learning, 2021).  The Hessian trace penalty proposed in this (review)  paper can reasonably be interpreted as an indirect way of controlling local curvature and hence local Lipschitz behaviour of the model, which is a well known route to adversarial robustness. Yet this line of work is not mentioned in the related work section and is not discussed in the analysis of the method. For the claims about novelty and effectiveness to be convincing, I believe the paper should
   1) explicitly acknowledge the Lipschitz regularization literature,
   2) conceptually explain how minimizing the Hessian trace differs from, or is related to, constraining network Lipschitz constants, and
   3) include empirical comparisons against representative Lipschitz regularized baselines (for example spectral norm or Lipschitz bound based methods as in Gouk et al.). Without this, it is hard to assess whether the observed robustness benefits are specific to the proposed Hessian scheme or simply an instance of a more general Lipschitz control effect that could be obtained in simpler and cheaper ways.

2. **Unclear computational complexity and practicality**

   The method relies on Hessian trace estimation with Hutchinson type probes, plus additional mechanisms such as layer sampling and gating. This suggests a nontrivial training overhead compared to standard SGD or even first order robust training, yet the computational complexity is not discussed in a systematic way. To support claims about the practicality of the approach, I would expect
   * explicit reporting of training time or compute (for example relative to standard training and to other robust baselines) on the main benchmarks,
   * a clear statement of the algorithmic complexity in big O notation as a function of number of parameters, number of layers, and number of probe vectors, and
   * if possible, a comparison of this overhead to that of alternative curvature based or Lipschitz based regularizers.
   At present, the reader is asked to accept that the method is tractable and scalable, but the evidence for this is lacking and not quantified, which weakens the practical part of the claim.

3. **Positioning and novelty of Hessian trace minimization**

   The paper presents Hessian trace regularization as a novel training strategy for improving robustness to bit flips. However, penalizing the trace of the Hessian using Hutchinson estimators has already been proposed in prior work for purposes such as improving generalization and finding flat minima, for example Liu et al., “Regularizing Deep Neural Networks with Stochastic Hessian Trace Estimation” and related “Hessian regularization of deep neural networks: A novel approach based on stochastic estimators of Hessian trace” works. The main novelty here is therefore the application and empirical study of this family of techniques in the specific context of robustness to weight bit flips, rather than the optimization objective itself. The current manuscript does not clearly acknowledge this and tends to present the Hessian trace penalty as a new training method. For the claims about contribution and evidence to be accurate, the authors should
   * more carefully situate their objective relative to existing Hessian trace regularizers and second order quantization methods,
   * explicitly frame their contribution as adapting and analyzing these ideas for bit flip robustness, and
   * ideally include at least one direct comparison to an existing Hessian trace based regularization method.
   Without this repositioning and comparison, the strength and originality of the methodological contribution, and therefore the interpretation of the empirical evidence, are overstated.

**Requested Changes:**

See Answer above

---

> ### Author Response · Authors · 2025-12-11
> **Thank You for the Constructive Feedback**
>
> We thank the reviewer for their time and valuable feedback. Below we provide our responses to all questions and concerns, along with the revisions we plan to incorporate into the manuscript.
>
> ----
>
> **Connection to Lipschitz Regularization**
>
> We acknowledge that Lipschitz-constrained training and curvature-based training share conceptual similarities. But we first want to clarify that controlling a model’s Lipschitz constant—i.e., its sensitivity to input-space perturbations—is fundamentally different from controlling the Hessian trace with respect to parameter-space perturbations, which are the primary cause of catastrophic failures under bit-flip attacks.
>
> Because Lipschitz constraints target input sensitivity, they are effective against adversaries manipulating the input (e.g., adversarial examples). However, they are known to be ineffective against bit-flip attacks, as demonstrated in prior work [1, 2] and confirmed by our own preliminary experiments. Prior work evaluated adversarial training [3]—an implicit means of reducing the Lipschitz constant—against bit-flip attacks and showed it is ineffective.
>
> One unexplored direction is explicitly bounding the model’s Lipschitz constant during training, such as via projected gradient descent [4]. In our preliminary attempts, enforcing Lipschitz bounds of {1, 10} on a ResNet-14 on CIFAR-10 leads to only 60–70% accuracy, which is impractically low. Moreover, training with explicit Lipschitz constraints (e.g., via spectral normalization) is computationally expensive, requiring roughly 10$\times$ more wall-clock time than our Hessian-aware training.
>
> Overall, we believe this discussion helps readers better contextualize our contributions. We will incorporate the above clarification, along with the prior work highlighted by the reviewer, into an expanded paragraph in the Related Work section, as part of our response to Reviewer 7ce3.
>
> ----
>
> **Computational Complexity and Practicality**
>
> We kindly remind the reviewer that Appendix E reports the  training overhead, where we provide an explicit comparison of wall-clock time between standard training and our Hessian-aware training. As noted there, our approach increases per-epoch wall-clock time by roughly 4–10$\times$ for smaller models such as those used in MNIST and CIFAR-10.
>
> But we also emphasize that we designed several mechanisms to address scalability while preserving effectiveness. In particular, our layer-sampling strategy leverages the empirical finding that bit-flip vulnerability is concentrated in the final fully-connected layers. By restricting Hessian-aware updates to only the last classification layer—rather than all layers—we reduce the overhead from 4–10$\times$ down to 1.18$\times$ on ImageNet-scale models, which we consider a practical cost.
>
> We also emphasize that our method aims to mitigate inference-time risks, where a single trained model may serve millions of queries. Accepting a modest one-time training overhead is therefore a favorable trade-off, especially since our defense incurs no inference-time overhead.
>
> Below is the Big-O complexity analysis we will add to Appendix E:
>
> **[Big-O Complexity]**
>
> Suppose
> $P$ is the number of parameters
> $L$ is the number of convolutional and fully-connected layers
> $p$ the number of Hutchinson probe vectors (i.e., the top-$p$ eigenvalue directions)
> $FLOPs_{l}$ is the number of floating-point operations (FLOPs) for layer $l$
>
> A standard forward-backward pass has complexity:
>
> $F_{SGD}  = \Theta \Big( \sum^{L}{FLOPs_{l}} \Big)$.
>
> A Hessian-vector product has the cost approximately comparable to an additional forward-backward pass, as follows:
>
> $F_{HVP}  = \Theta \Big( \sum^{L}{FLOPs_{l}} \Big)$.
>
> Because our algorithm performs one standard SGD for the loss and $p$ HVPs per iteration, the FLOPs per training step are:
>
> $F_{step} = \Theta \Big( F_{SGD} + p \cdot F_{HVP} \Big) = \Theta \Big( (1 + p) \sum^{L} FLOPs_{l} \Big)$.
>
> Note that the sum of FLOPs$_{l}$ linearly scales with the number of parameters $P$, then:
>
> $F_{step} = \Theta \Big( ( 1 + p ) P \Big)$.
>
> Under our layer-sampling optimization, we restrict Hessian computations to only the last layer $l’$. In this case,
>
> $\sum^{L}{FLOPs_{l}} \rightarrow FLOPs_{l'}$
>
> Which reduces the training-time overhead to only 1.18$\times$ of standard training in our ImageNet experiments.
>
>
> **[Comparison to Alternatives]**
>
> Many Lipschitz regularizers [4] require computing per-layer spectral norms using power iterations, which also introduce $O(k \cdot P)$ overhead per step, where $k$ is the number of power iterations. Sharpness-Aware Minimization (SAM) similarly incurs the cost of two forward-backward passes per update $O(2P)$.
>
> In comparison, our approach—especially with layer-sampling—achieves a competitive or smaller computational overhead while providing resilience against bit-flip attacks.
>
> ----
>
> (cont'd in the next comment)

---

> > ### Author Response · Authors · 2025-12-11
> > **Thank You for the Constructive Feedback - Cont'd**
> >
> > **Novelty and Positioning of Hessian-trace Minimization**
> >
> > We thank the reviewer for pointing out prior work on Hessian-trace regularization for improving generalization and finding flat minima. Our contribution is *not* to propose a new optimization objective; rather we will clarify that our novelty lies in the following aspects:
> >
> > 1. Demonstrating that Hessian-trace regularization directly targets the root cause of catastrophic bit-flip failures: extreme curvature with respect to parameter-space perturbations. Prior trace-regularization work optimizes for generalization or flat minima, but did not study or motivate the method in the context of weight-bit perturbations.
> >
> > 2. Analyzing the connection Hessian curvature and bit-flip vulnerability. Existing work does not consider parameter corruption, and no prior work investigates whether such second-order methods improve robustness to hardware-level bit-flip attacks.
> >
> > 3. Adapting and redesignoing the training procedure spe thecifically for bit-flip robustness: proposing top-$p$ directional curvature since top-1 direction is insufficient for improving bit-flip resilience and introducing layer-sampling based on empirical layer-wise sensitivity for efficient scaling to ImageNet. These modifications are not present in prior Hessian-trace works and are crucial for making this approach effective at scale.
> >
> > 4. Providing an extensive empirical study showing that second-order curvature control is an effective defense against both random and adversarial bit-flip attacks, a setting absent from the prior literature.
> >
> > We will revise the manuscript to more clearly as follows to acknowledge this line of research:
> >
> > 1. Reposition the method clearly as an adaptation of stochastic Hessian-trace regularization for the $new$ purpose of bit-flip robustness, and
> >
> > 2. Add a paragraph in Related Work that clearly describes the connection to prior Hessian-trace regularizers and second-order methods.
> >
> > ----
> >
> > Thank you again for their thoughtful and constructive feedback. We will be happy to address any additional questions and will work on revising the manuscript to incorporate all suggested improvements.

---

### Decision · Action_Editor_QXE5 · 2026-03-16

**Recommendation:** Accept as is

**Audience:**

Yes

**Audience Explanation:**

The work sits at the intersection of robustness, optimization and systems, all of which are sizeable parts of the TMLR audience.

**Claims And Evidence:**

Yes

**Claims Explanation:**

The reviewers felt that the manuscript supports the claims reasonably well but had several clarifying questions for the authors.  After responses by the authors, reviewers agreed that the claims are well supported, but we would encourage the authors to incorporate some of that justification into the final version of the paper.